# Ultra-sensitive nanometric flat laser prints for binocular stereoscopic image

Dejiao Hu[1,4], Hao Li[1,4], Yupeng Zhu[2], Yuqiu Lei[1], Jing Han[1], Shilin Xian[2], Jiajin Zheng[3], Bai-Ou Guan[1], Yaoyu Cao[1✉], Lei Bi [2✉] & Xiangping Li [1✉]

Two-dimensional (2D) transition metal dichalcogenides (TMDs) with tantalizing layer-dependent electronic and optical properties have emerged as a paradigm for integrated flat opto-electronic devices, but their widespread applications are hampered by challenges in deterministic fabrication with demanded shapes and thicknesses, as well as light field manipulation in such atomic-thick layers with negligible thicknesses compared to the wavelength. Here we demonstrate ultra-sensitive light field manipulation in full visible ranges based on $MoS_2$ laser prints exfoliated with nanometric precisions. The nontrivial interfacial phase shifts stemming from the unique dispersion of $MoS_2$ layers integrated on the metallic substrate empower an ultra-sensitive resonance manipulation up to 13.95 nm per $MoS_2$ layer across the entire visible bands, which is up to one-order-of-magnitude larger than their counterparts. The interlayer van der Waals interactions and the anisotropic thermal conductivity of layered $MoS_2$ films endow a laser exfoliation method for on-demand patterning $MoS_2$ with atomic thickness precision and subwavelength feature sizes. With this, nanometric flat color prints and further amplitude-modulated diffractive components for binocular stereoscopic images can be realized in a facile and lithography-free fashion. Our results with demonstrated practicality unlock the potentials of, and pave the way for, widespread applications of emerging 2D flat optics.

[1] Guangdong Provincial Key Laboratory of Optical Fiber Sensing and Communications, Institute of Photonics Technology, Jinan University, Guangzhou, China. [2] National Engineering Research Center of Electromagnetic Radiation Control Materials, University of Electronic Science and Technology of China, Chengdu, China. [3] College of Electronic and Optical Engineering, Nanjing University of Posts & Telecommunications, Nanjing, China. [4] These authors contributed equally: Dejiao Hu, Hao Li. ✉email: yaoyucao@jnu.edu.cn; bilei@uestc.edu.cn; xiangpingli@jnu.edu.cn

Since the first discovery[1,2], two-dimensional (2D) materials with exceptional optical and electronic properties have offered an unprecedented platform for developing ultra-compact flat opto-electronic devices with a considerable degree of miniaturization. Transition metal dichalcogenides (TMDs)[3,4] exhibiting sensitive layer-dependent properties, including indirect-to-direct bandgap transition from bulk states to monolayers, emerge as a peculiar complementary to graphene for investigating excitonic light-matter interactions. As such, onrushing developments of extensive applications have been witnessed in photodetectors[5,6], valley emissions[7], transistors[8,9], and memories[10,11].

In spite of manifesting enticing high refractive indices[12,13], the light field manipulation in nanometric TMD layers remains dull sensitivity to their vanishing small thickness compared to the wavelength of operation. Until very recently, nanostructured TMD layers at elevated thicknesses to hundreds of nanometres are starting to be appreciated as high-index dielectric resonators supporting distinct geometry-dependent Mie resonances, which starts a research to shaping light's wavefronts by 2D materials based optical elements[14,15]. However, these demonstrations are achieved at the cost of degraded device compactness and integration. Moreover, the fabrication of nanostructured TMD layers mainly relies on mechanical or chemical exfoliation from bulk materials[16,17] and subsequently follows complex lithography procedures to produce demanded shapes and thicknesses[14,15]. Even though these approaches are demonstrated effective for fundamental researches, aforementioned challenges remain the major hurdle for the pragmatic and widespread applications of the emerging 2D flat optics.

Here, we demonstrate ultra-sensitive light field manipulation in resonance spectra by nanometrically maneuvering thickness of MoS$_2$ layers through a laser exfoliation technique. The anisotropic thermal conductivity of layered MoS$_2$ films integrated on a metallic substrate with two orders of magnitude difference in in-plane and out-of-plane directions leads to a giant temperature gradient along stacking layers. Combing the interlayer van der Waals interactions, it enables layer-by-layer laser exfoliation effects[18,19] which provide a pragmatic and lithography-free means to on-demand structure MoS$_2$ flakes at atomic layer precisions and subwavelength feature sizes. By introducing the giant interfacial phase shifts associated with imaginary parts of their refractive indices, the resonance exhibiting ultra-sensitivity to the nanometric thickness can lead to not only vivid color appearances in full visible ranges but also large reflectance amplitude modulations. The proof-of-principle demonstration of flat color prints and further amplitude-modulated diffractive components for multi-perspective stereoscopic images unfolds the potential of 2D flat optics with practicality and up-scalability.

## Results

**Ultra-sensitive resonance manipulation through laser exfoliating nanometric MoS$_2$ layers.** The typical configuration of nanometric MoS$_2$ multilayers integrated on a gold substrate for ultra-sensitive resonance manipulation by direct laser writing technique[20,21] is schematically illustrated in Fig. 1. The MoS$_2$ thin films prepared through pulsed laser deposition with an initial thickness of 20 nm were deposited on a gold substrate (see "Methods"). The layered crystal structure of as-grown MoS$_2$ thin films was verified by both X-ray diffraction (Supplementary Fig. 1) and high-resolution transmission electron microscopy (Supplementary Fig. 2) experiments. The optical constants from the ultraviolet to near-infrared wavelength range were obtained using spectroscopic ellipsometry (Supplementary Fig. 3), which are used for calculations throughout the paper. A continuous

wave (CW) laser beam at the wavelength of 532 nm was focused by an objective lens (×50, NA = 0.75) to pattern MoS$_2$ layers. Light absorption in the upper layers can produce a local temperature rise that burns out atoms in the vicinity of the focal region, which can be dexterously controlled by the laser recipe. Figure 1b shows the reflection optical image of a tangram pattern fabricated through the laser writing method. The zones with distinct color appearances represent MoS$_2$ thin films with different nanometric thicknesses, verified by the topographic atomic force microscopy (AFM) image. Indeed, the structure exhibits extreme sensitivity in resonance tunability across in the full visible range through subtle variations of the thickness of MoS$_2$ thin films by merely tens of atomic layers.

To gain insights into the ultra-sensitive resonance manipulation, we first review the Fabry-Perot (FP) resonance support by a thin film. FP resonance modes are standing waves formed by light wave propagating back and forth between two reflective interfaces, where constructive interference occurs after the light wave travels a round-trip. The resonance condition can be matched once the total phase accumulation, including both propagation phase shifts and interfacial phase retardations acquired at interfaces to be an integer modulus of 2π. Thus, the resonance wavelength can be simply derived as

$$\lambda = \frac{2n}{m - \left(\frac{\varphi_1 + \varphi_2}{2\pi}\right)} h \qquad (1)$$

where $n$ is the real part of the refractive index of the FP layer, $h$ is its thickness, $m$ is the order of resonance, and $\varphi_1$ and $\varphi_2$ are interfacial phase shifts defined to lie in the interval [0, 2π] at the two interfaces, respectively. It can be seen that the sensitivity of the resonance wavelength to the FP thickness is mainly governed by the refractive index of the material and total interfacial phase shifts at the two interfaces. Especially, when the total interfacial phase shifts are reaching 2π, the denominator will approach zero for the first order resonance ($m = 1$). In this case, the resonance wavelength will be ultra-sensitive to even nanometric thicknesses of the FP medium, which leads to performances superior to conventional optical coatings relying on the propagation phase accumulation in a quarter-wave-thick film[22] (Supplementary Note and Supplementary Fig. 4).

To realize such ultra-sensitive resonance manipulation, dielectric thin films with large complex indices as well as proper substrate designs are of vital importance. The unique dispersion of MoS$_2$ thin films integrated on the Au substrate empowers broadband interfacial phase shifts reaching 2π in the visible regime (Fig. 1c). Consequently, the largest tunability in reflectance spectra corresponding to a wide color palette by varying nanometric thickness of MoS$_2$ layers can be realized, which exceedingly outperforms the other substrates such as Ag and Silicon (Supplementary Fig. 5). Figure 1d depicts the theoretical calculation results of the reflectivity spectra of the MoS$_2$–Au structure with different numbers of layers (see "Methods"). As can already be inferred from the white dotted line marking out the evolution of resonance wavelengths, nearly 400 nm shift from 800 nm to 430 nm covering the entire visible light can be obtained by a thickness variation from 30 layers to a monolayer. It is worth noting that even the extinction-associated interfacial phase shifts[23] are ubiquitous among dissipative materials with a large imaginary part of refractive index, the enticing dispersion properties of MoS$_2$ thin films covering the outermost range in the complex index diagram (Fig. 1c) manifest a far superior sensitivity (defined as resonance wavelength shifts divided by the number of layers) of 13.95 nm per MoS$_2$ layer, which shows up to 6 times and 2 times greater than that of 2.3 nm

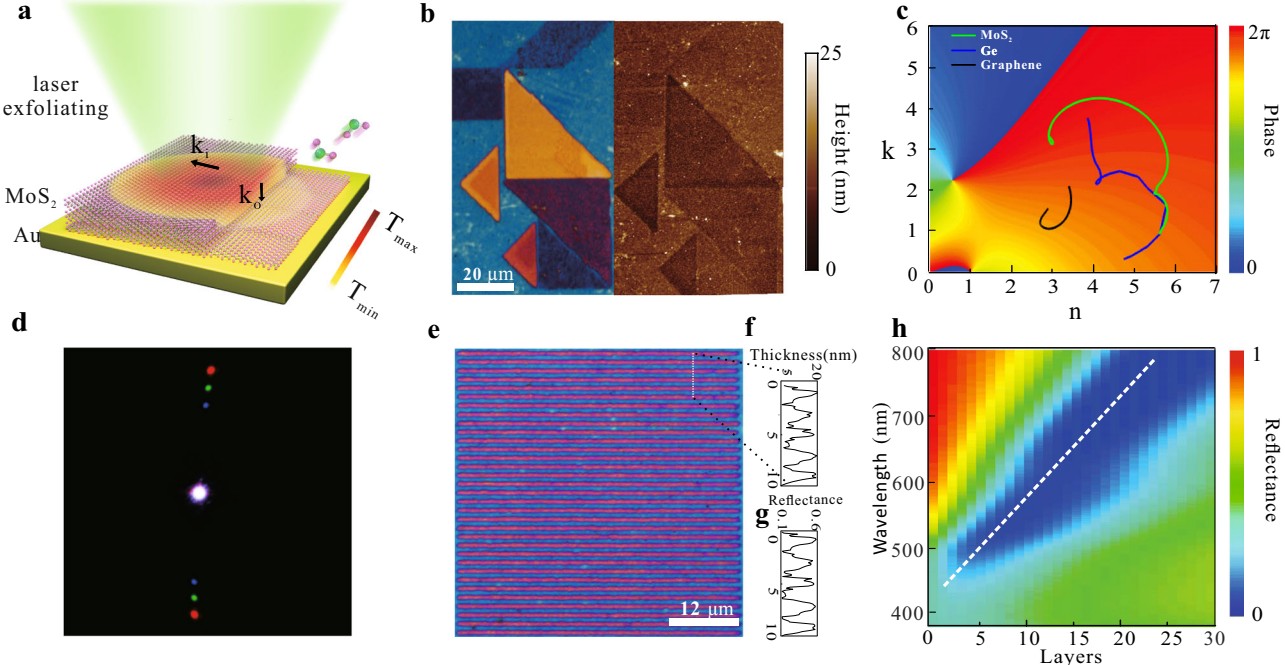

**Fig. 1 Ultra-sensitive resonance manipulation by laser exfoliating MoS$_2$ thin films. a** Schematic diagram of tightly focused laser beams for exfoliating multilayer MoS$_2$ integrated on the Au substrate with atomic thickness precisions. **b** Optical and the AFM images of a Tangram pattern printed by the facile laser writing method. **c** The calculated total interfacial phase shifts for a dielectric layer-Au configuration with variant complex refractive index overlaid with the complex refractive index diagram of MoS$_2$ (green), graphene (black), and Ge (blue). **d** The calculated reflectance spectra of the MoS$_2$ thin films on the Au substrate with different numbers of layers. **e** Microscopic image of the grating structure fabricated with a period of 1800 nm. **f** Surface corrugation curve of patterned nanometric gratings corresponding to the dotted line. **g** Reflectance amplitude modulations by such grating structures. **h** Diffraction image retrieved by using three beams at wavelengths of 450, 550, and 650 nm simultaneously.

per graphene layer and 5.9 nm per Ge layer, respectively (Supplementary Fig. 6).

The ultra-sensitivity of resonances to the nanometric thickness can lead to not only full visible spectral tuning but also large reflectance amplitude modulations. Close to the resonance wavelength of 532 nm, a 5 nm-thick MoS$_2$ film corresponding to 7 atomic layers can lead to nearly vanishing reflectance, while a 20 nm-thick film can shift the resonance away, leading to an enhanced reflectance approaching 0.5. As such, laser patterned nanometric flat grating structures (Fig. 1e) with a surface corrugation of 15 nm (Fig. 1f) are capable of light field manipulation based on strong amplitude modulation strength up to 0.44 (Fig. 1g). As an illustration, the diffraction patterns under 450, 550, and 650 nm lasers simultaneous illumination unambiguously verify the light field manipulation by such nanometric flat gratings (Fig. 1h).

Apparently, the interlayer interaction by van der Waals forces results in the dominant heat dissipation along the in-plane direction rather than the out-of-plane direction[18,24]. The layered MoS$_2$ films on the Au substrate exhibit two orders of magnitude difference in thermal conductivity along the in-plane and out-of-plane directions, which can result in a giant temperature gradient along stacking layers (Supplementary Fig. 7). The local temperature in the upper layers is quickly built up to exceed the burning temperature and the beneath layers are less susceptible unless irradiation at high powers. This provides the basis for reliable thickness control at atomic layer precisions by establishing a standardized laser exfoliating recipe. Figure 2a–d illustrate the systematic investigation of nanometric thickness control at variant laser doses, and characterization obtained through AFM and Raman spectroscopy (see "Methods"). The laser scanning speed is optimized and fixed at 0.1 mm/s in the whole experiment (see Methods). By varying the laser irradiance powers with an

increment of 8 mW, the thickness of MoS$_2$ thin films can be thinned in a staircase behavior with a step height of approximately 3 nm corresponding to 4 layers (given the monolayer thickness about 0.67 nm[25]). The surface roughness is on the order of 7.491 nm due to the presence of splashed or unremoved residuals, albeit no significant influence on their light field manipulation capabilities.

Since the frequency interval between $E_{2g}$ and $A_{1g}$ Raman modes depends monotonically on the number of MoS$_2$ layers[26,27], Raman spectroscopy is utilized to characterize the laser exfoliation process. Figure 2c depicts the evolution of Raman spectroscopy during the laser exfoliation. Initially, the frequency interval is 26 cm$^{-1}$ for the 20 nm thick flakes indicating a multilayer state. As the laser dose increases, the frequency interval monotonically decreases close to 21 cm$^{-1}$ implying a monolayer to bilayer state. The above characterization consolidates the effectiveness of the laser writing technique for thickness control at atomic precisions as well as up-scalability for on-demand fabricating layered MoS$_2$ based optical elements.

From Eq. (1), it clearly reveals that the nanometric thickness variation of MoS$_2$ thin films integrated on the Au substrate leads to an ultra-sensitive light field management in resonance wavelengths. Figure 2e and f show the theoretical calculation results of reflection spectra from that structure at variant nanometric thicknesses of MoS$_2$ thin films as well as corresponding experimental results. The experiments are in good congruence with the theoretical predictions. It can be seen that the reflection valley caused by the resonant absorption can be continuously tuned in the whole range of visible light from the wavelength of more than 700 nm to 500 nm, by merely a thickness decrease from ~20 nm to 5 nm. Optical micrographs of the colors generated from the laser thinned zones and corresponding color coordinate map are shown in Fig. 2g, h, respectively.

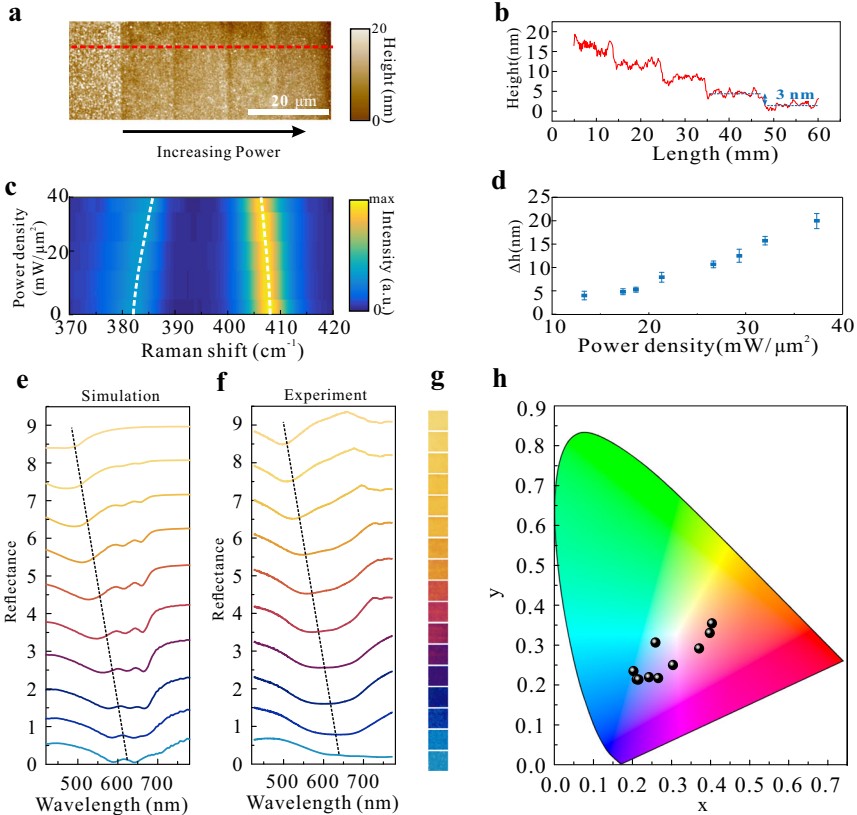

**Fig. 2 Laser exfoliating MoS$_2$ thin films with nanometric thickness precisions. a** The AFM image of laser thinned regions obtained at different laser doses. **b** The height change corresponding to the red dotted line in (**a**). The height of the steps is about 3 nm. **c** The variation of characteristic Raman spectra of MoS$_2$ flakes as the increase of laser powers. **d** Extracted thickness change from Raman spectra at corresponding laser powers. **e, f** Reflection spectra of MoS$_2$ thin films prepared on the Au substrate with different nanometric thicknesses. **g, h** Experimentally obtained reflection color palettes and color coordinate diagram through laser exfoliating at variant powers.

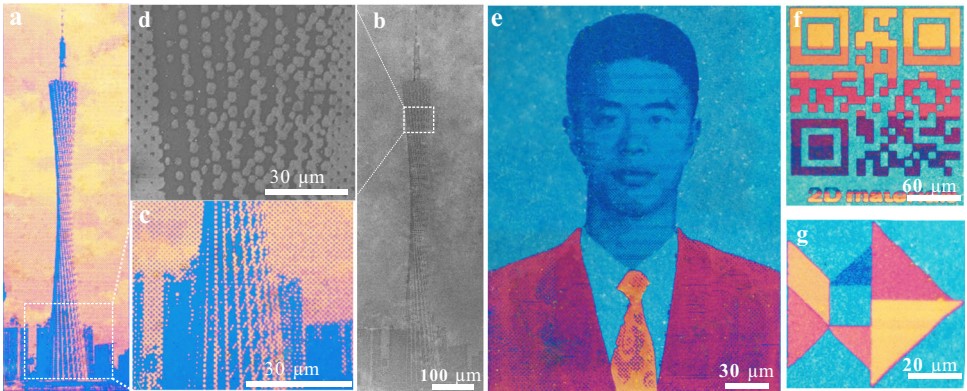

**Fig. 3 High fidelity color images by laser exfoliated nanometric flat prints. a, c** Are optical microscopic image and zoom-in view. **b, d** Are SEM image and sectional zoom-in view. A collection of color images printed by the laser exfoliation method with atomic thickness precisions and subwavelength feature sizes in continuous tone (**f, g**) and half tone modes (**a, e**) demonstrates the capability of high color fidelity and high spatial resolution.

**Flat prints and binocular stereoscopic images.** The ultra-sensitive resonance control allows the generation of nanometric flat prints for color images[28–31] (Supplementary Fig. 8). Figure 3a–e are collections of halftone images patterned by laser writing methods. The color halftone effect is realized through controlling the printed pixel density. The underlying mechanism of laser exfoliation can be attributed to photothermal effects, where the local heating effect and heat diffusion can be flexibly controlled by the focusing conditions to pattern subwavelength feature sizes. By dexterous control the laser recipe, the patterned

image composed of subwavelength scale pixels can achieve a high spatial resolution up to 58,000 dpi. The minimum pixel size is measured around 400 nm given a focusing lens with a numerical aperture of 0.75. To demonstrate the up-scalability, a *Canton Tower* with a millimeter scale was printed with a high resolution and high fidelity (Fig. 3a). Scanning electron microscope (SEM) image of the prints is depicted in Fig. 3d. The SEM image and optical micrograph of the selected region with fine details are shown in Fig. 3b, c, respectively. Figure 3f–g show another example of images or even color QR codes printed with

continuous tone. More examples of color images generated by nanometric flat pigments can be seen in Supplementary Fig. 9. Because the ultra-sensitive resonance stems from interfacial phase shifts which are immune to the incident angle, the color images demonstrate an angle robust performance (Supplementary Fig. 10). Detailed performance was further compared with other structural color literature in Supplementary Table 2.

In addition to sensitive resonance wavelength modulation to generate flat color prints, this nanometric thickness dependent resonance control opens up efficient diffraction effects based on strong amplitude modulations to shape the impinging wavefronts. Supplementary Fig. 11 illustrates the measured diffraction efficiency of patterned micrograrings with a period of 1000 nm and a thickness corrugation of 10 nm as a function of wavelengths. Given the ultra-thin nanometric thickness of 20 nm of the top $MoS_2$ thin films, the first-order diffraction efficiency (defined as the first-order intensity divided by summed intensities of first-order and zeroth-order) can reach up to 36% at the optimal wavelength of 560 nm, which is sufficient to develop a binocular stereoscopic view approach by multi-perspective diffractive images. The principle and the configuration are schemed in Fig. 4a, b. Two perspective images projected at two different directions were encoded into two sets of amplitude-modulated diffractive gratings with different periods of 1200 nm and 1800 nm, respectively (Supplementary Fig. 12). By horizontally interleaving the two sets of laser printed gratings to form a 10 μm sized "pixel", two perspective images can be diffracted and projected to different directions corresponding to left and right eyes at an oblique incidence (Fig. 4b). Figure 4c, d show the captured portrait images taken from different perspective that can form a binocular stereoscopic view. The zoom-in views of the SEM image and optical micrograph of the laser printed interleaved grating structures are shown in Fig. 4e, f.

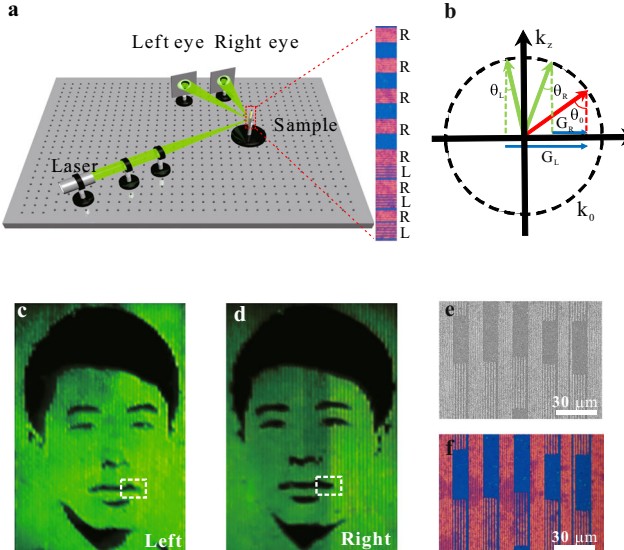

**Fig. 4 Laser exfoliated nanometric flat diffractive prints for binocular stereoscopic images. a** The optical configuration for binocular stereoscopic images. The inset shows optical microscopic images of laser exfoliated nanometric flat diffractive pixels made of two sets of grating structures. **b** Design principle of stereoscopic views by diffracting light from binocular positions ($\theta_L$ and $\theta_R$) at oblique incident angles ($\theta_O$). The dashed circles with radius $k_0 = 2\pi/\lambda$ represent the wave vector of light. **c**, **d** are different images acquired by the left and right eyes, which were false-color images of the human face taken from different angles, respectively. **e**, **f** Enlarged views of optical microscopic image and corresponding SEM image in the white dotted boxes in (**c**, **d**).

## Discussions

By removing complex and sophisticated electron beam nano-lithography procedures that are heavily resorted to, the demonstrated laser writing methods with atomic thickness precisions and subwavelength feature sizes paves the way to on-demand fabrication of ultra-thin TMD based opto-electronic devices with great facility and up-scalability. By introducing the interfacial phase shifts, the exotic dispersion of TMD materials empowers an ultra-sensitive light field manipulation scheme and opens avenues to a class of nanometric flat optical elements. Furthermore, the proof-of-principle demonstration of flat color prints and further multi-perspective stereoscopic views base on such nanometric amplitude-modulated diffractive pixels provides a viable scheme to construct light field manipulation based flat optics with ultra-compact footprints and superior miniaturization. Combing their layer-dependent behaviors of bandgaps, the ultra-sensitive light field management in resonance control by the nanometric thickness of TMD thin films can dramatically push the state-of-the-art and nourish multifarious functionalities of emerging 2D integrated opto-electronic devices.

## Methods

**Sample preparation.** The $MoS_2$ thin film with a lateral size of several millimeters was firstly deposited onto $Si/SiO_2$ (300 nm $SiO_2$) substrate with pulse laser deposition (PLD) method, and then was transferred onto a gold substrate obtained by heat evaporation. In the PLD process, a KrF laser with 248 nm wavelength was used and the commercial $MoS_2$ target. The condition of fabrication $MoS_2$ layer was $10^{-5}$ pressure and 780 degree. After the deposition, the pressure was maintained and annealed for 5 min at the same condition, and then the temperature was decreased at a rate of 10 °C/min to the room temperature. The power of the laser pulse was 150 mJ with a repeat frequency of 5 Hz. The thickness of the $MoS_2$ layer was controlled by the pulse number of the laser. The $MoS_2$ film was transferred onto the gold substrate with wet transfer method, where a polymethylmethacrylate (PMMA) layer was spinned onto the $Si/SiO_2/MoS_2$ films with a spin rate of 500 rmp for 10 s and then 3000 rpm for 60 s. And then, the PMMA layer was immobilized through baking for 10 min at a temperature of 100 °C. After this, the sample was immersed in KOH solution to remove the substrate and the $MoS_2$ with PMMA film float onto the solution surface. Then the residual KOH was washed by putting the $MoS_2$ and PMMA film in deionized water for several seconds. The $MoS_2$ and PMMA film was then spread onto the gold substrate and baked for 10 min under a temperature of 95 °C. Then the PMMA layer was removed by immersing in acetone, ethanol and deionized water for 30 s, respectively.

**Laser printing.** The samples were placed on a computer-controlled 3D translation stage. The continuous wave laser beam at the wavelength of 532 nm was attenuated and focused by an objective lens (×50, 0.75 NA). The effective exposure time is 4.33 ms. The beam power was adjusted to the desired value with a neutral density attenuator. During the laser printing, the movement of the translation stage was synchronized with the laser shutter to control the exposure of irradiances. The authors affirm that human research participants provided informed consent for publication of the images in Figs. 3, 4, and S9.

**Sample characterization and optical measurements.** Reflection color images of the sample were characterized using an objective lens (MPlanFL N, ×50×/0.8, Olympus Co.). A CCD camera (Olympus, BX53, Olympus Co) was used to acquire the images from the sample. The spectra were characterized with a home built confocal microscope coupled to a spectrometer (Andori500). The sample was illuminated using a halogen white light source using an objective lens (MPlanFL N, ×20×/0.45, Olympus Co.). The reflected light was collected through the same objective lens and recorded using a spectrometer. The reflected intensity was normalized by the spectrum of the lamp obtained by reflection measurements with a silver mirror.

**Characterization.** An atomic force microscope (Ntegra solaris, NT-MDT Spectrum Instruments, Moscow, Russia) has been used to study the topography and determine the height of patterned $MoS_2$ flakes. A Raman spectrometer (RENISHAW inVia) was used in a backscattering configuration excited with a visible laser beam (λ = 532 nm, power 5 mw) to confirm the layer number of $MoS_2$ flakes.

The layered structure is verified by X-ray diffraction (XRD) and high-resolution transmission electron microscopy (TEM) experiments. The sample is a cross-section sample cut with FIB (FEI Quanta 3D FEG). TEM images and Local energy-dispersive X-ray spectroscopy (EDX) was carried on a FEI Tecnai F20 S-TWIN operated at 200 kV. High-resolution ADF-STEM were performed in a probe-corrected STEM (FEI Titan Chemi STEM) operated at 200 kV. For illumination

and in situ fabricatior, a convergence angle of 21.4 mrad, a probe current of ~70 pA, a range of acceptance angle of ADF detector was 43.4–200 mrad, and a pixel dwell time of 10 μs.

**Numerical calculations**. The reflection and transmission coefficients from a single interface is calculated by using the Fresnel equations, $r_{ij} = (p_i - p_j)/(p_i + p_j)$, $t_{ij} = 2p_i/(p_i + p_j)$, $p_i = n_i \cos(\theta_i)$. Here, $t_{ij}$ and $r_{ij}$ are transmission and reflection coefficients under illumination from medium $i$ to medium $j$, $n_i$ is the complex refractive index of medium $i$ and $\theta_i$ is the angle between the propagating direction of the light wave within the medium and the normal direction of the layered films. The reflection phase shift from the interfaces was extracted from the reflection coefficients and then were added up to the total phase shift. The reflection from the air-FP-substrate layers was directly calculated by using $r = r_{12} + \frac{t_{12}t_{21}r_{23}e^{-i2\varphi}}{1 - r_{21}r_{23}e^{-i2\varphi}}$, where 1, 2, 3 represent the air, FP layer, and substrate, respectively, $\varphi$ is the propagation phase accumulation in the FP layer.

## Data availability

The data that support the plots within this paper and other findings of this study are available from the corresponding author upon reasonable request.

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

## Acknowledgements

The authors are grateful for the support of STEM sample fabrication and characterization by Ms. Li Gu and Prof. Chuanhong Jin in Zhejiang University. This research was supported by National Key R&D Program of China (2018YFB1107200), National Natural Science Foundation of China (NSFC) (Grant 61705084), Guangdong Provincial Innovation and Entrepreneurship Project (Grant 2016ZT06D081).

## Author contributions

D.H. and H.L. contributed equally to this work. X.L. and B.L. conceived the idea and supervised the project. D.H., H.L., and Y.L. performed the experiments. H.L. and H.J. assisted spectral measurements. Y.Z. and S.X. prepared the sample. D.H. and H.L. performed the modelling and simulation. D.H., H.L., J.Z., B.G., B.L. Y.C., and X.L. analyzed data and prepared the manuscript. All authors contributed to the discussion and manuscript writing.

## Competing interests

The authors declare no competing interests.
