## [Peer Review File · Nature Communications]

Reviewers' comments:

Reviewer #1 (Remarks to the Author):

Li et al., demonstrate colors from MoS₂ films of only a few layers on a gold substrate, aided by control over the film thicknesses with a laser exfoliation process. The paper is interesting with nice demonstrations of stereoscopic imaging from two types of ultrathin TMDs gratings, but the main idea of the paper—that colors can be achieved from nanometer-thick lossy dielectrics on metal has already been reported in [Mikhail A. Kats et al., Nature Materials, 12 20 (2013)]. Instead, Li et al., use MoS₂ with gold, which appears to produce a wider color tunability range due to interfacial phase shifts amounting to 2π , and demonstrate thickness control using a laser exfoliation process. Because these efforts can be considered only incremental compared to the 2013 Nature Materials paper by M.A. Kats et al., I believe this paper is better suited for another journal. Areas requiring major clarification are also described below.

1. Equation 1 does not appear to be universal in that it does not reproduce the commonly known interference equation for a lossless dielectric on a metal substrate. For such a case the total interfacial phase shift is 2π , and the constructive interference condition would be $m \times \lambda = 2n$. Equation 1, however, would show a singularity for such a case.
2. The theory behind the laser exfoliation process needs to be clarified more in detail. The depth of focus of the focused laser beam will cover the entire film thicknesses, but the authors seem to suggest that heating occurs from the top layer. I believe that the gold substrate plays a significant role in the thermal heating process and the overall thermal profile of the film that the authors are describing, but its role is not mentioned in the manuscript.
3. The laser power density, not the absolute power, should be reported. A discussion of the damage to the film should be included as this can significantly modify the optical properties of the film.
4. The captions and figures require improved labeling in the Supplementary section. For example, the 'efficiency' labeled in Figure S6 needs to be specified. Which efficiency is this?

Reviewer #2 (Remarks to the Author):

The manuscript used laser ablation/sublimation to reduce thickness of a bulk MoS₂ layer to change Fabry-Perot resonance to change color of the film. FP resonance based color generation is most widely known and used phenomenon. Many materials are patterned following conventional lithographic techniques as well as lasers to form FP color. There are many publications on this topic. My few specific questions/comments are:

- What is the film quality of the PLD grown MoS₂ layer? It will have many defects and will be a poor polycrystalline film. I don't see any justification for calling it a 2D material in this context. Here the laser is doing bulk micromachining of the material.
- The manuscript will benefit if authors show film characterizations, electrical as well as optical.
- Angular dependency of the color should be studied for any useful application.

Reviewer #3 (Remarks to the Author):

This manuscript describes a simple and efficient method to exfoliate MoS₂ and also shows fidelity color images and a binocular stereoscopic image by irradiating continuous laser to multi-layer

MoS₂ on the Au substrate as applications. Indeed, I believe this method can extend the potentials of 2D materials to be used in more practical applications. The manuscript is clearly written, contains noble results, and deserves to be acceptable in Nature Communications.

However, I feel that some minor points in the manuscript need to be examined.

1. It is not explained why the gold substrate is better than aluminum and silicon substrate. Because of material loss? or plasma frequency? Is it normal for any TMD materials? Please explain shortly why.
2. Figure 1c is not easy to understand. What is the main purpose of this graph? To notice broadband interfacial phase shifts? Where is the region 2π phase shifts? The red color region? or the phase is abruptly jumped (where n is 2, k is 3.5)? What is the k in Figure 1c? Extinction coefficient?
3. It is not distinguishable well between MoS₂ and graphene, and Ge and background phase color due to similar colors in Figure 1c. Please change the colors to distinguish well. I cannot understand why the calculated results of Ge and graphene are suddenly shown without having an explanation. These results make me confused as well.
4. The authors mentioned and compared the sensitivity of graphene, Ge, and MoS₂. However, it is not clear the definition of sensitivity and how the number (12.83 nm) could be obtained. Please explain more details to be understandable.
5. Line 96,137, 166, and 208, "showcases"  shows?
6. Figure 2 describes the exfoliated MoS₂ film thickness depend on laser power. However, the exposure time of the laser is not explained.

#####

Reviewer #1 (Remarks to the Author):

Li et al., demonstrate colors from MoS₂ films of only a few layers on a gold substrate, aided by control over the film thicknesses with a laser exfoliation process. The paper is interesting with nice demonstrations of stereoscopic imaging from two types of ultrathin TMDs gratings, but the main idea of the paper—that colors can be achieved from nanometer-thick lossy dielectrics on metal has already been reported in [Mikhail A. Kats et al., Nature Materials, 12 20 (2013)]. Instead, Li et al., use MoS₂ with gold, which appears to produce a wider color tunability range due to interfacial phase shifts amounting to 2π, and demonstrate thickness control using a laser exfoliation process. Because these efforts can be considered only incremental compared to the 2013 Nature Materials paper by M.A. Kats et al., I believe this paper is better suited for another journal. Areas requiring major clarification are also described below.

Reply: We thank the reviewer for acknowledging the quality of our work. However, we cannot agree with the reviewer that our work is incremental to the referenced NM paper (Nature Materials, 2013). In this work, we demonstrated a new scheme of nanometric flat laser prints exhibiting ultra-sensitive atomic-layer-dependent resonances that allow both full-visible color images and binocular stereoscopic image by a simple layer-by-layer laser exfoliation technique, representing the first of its kind. Although the two works share the interfacial phase shifts and the associated color effect in common, we reinforce that the innovation of this work making it different from the referenced NM paper lies in the following facts.

- 1) Firstly, the binocular stereoscopic image is based on diffraction effects by such nanometric flat prints which is different from the color images reported in the referenced NM paper based on reflection. The ultra-sensitive layer-dependent resonances stemmed from interfacial phase shifts cannot only lead to vivid color appearances but also large amplitude modulations of reflectance upto 0.44 in laser printed binary grating structures with a surface corrugation of merely 15 nm. The demonstration of binocular stereoscopic images based on interleaved pixel design made of two sets of nanometric flat laser prints opens the potential for planar diffractive optics, which is different from and out of scope of the NM paper.
- 2) Secondly, the anisotropic thermal conductivity of layered MoS₂ films deposited on the metal substrate enables a layer-by-layer laser exfoliation method for thickness control of laser prints with atomic accuracy. Even the initial thickness of the layered film of 20 nm is within the focal depth (~ 1.3 μm) of the tightly focused beam, it allows precise exfoliation of atop atomic layers without adversely affecting the bottom layers. Thanks to the two-order-of-magnitude difference in thermal diffusion coefficients of layered MoS₂ films between the lateral and vertical directions as well as a heat sink effect by the metal substrate, a sharp temperature gradient of 120 degrees/nm along stacking layers can be achieved to allow laser exfoliation with a demonstrated thickness step of 3 nm corresponding to 4 atomic layers at each laser sweeping. This top-down laser writing method with atomic precision opens the possibility for in-situ thickness control for laser prints after preparation, which makes it fundamentally different from the case where isotropic Ge thin film was prepared in lithography without the post-fabrication capability.

Please see corresponding revisions in the abstract, introduction, main text (Page 2, Page 4, Page 7) as well as Figure 1.

Comment 1) Equation 1 does not appear to be universal in that it does not reproduce the commonly known interference equation for a lossless dielectric on a metal substrate. For such

a case the total interfacial phase shift is 2π , and the constructive interference condition would be $m \times \lambda = 2 n h$. Equation 1, however, would show a singularity for such a case.

Reply: We thank the reviewer for his/her professional comment. The Eq. 1 is indeed universal to both lossy and lossless media with the interfacial phase shifts defined to lie in the interval $[0, 2\pi)$. For lossless media, the interfacial phase shifts are either 0 or π depending on the difference in refractive indices of the two materials. Regarding the lossy material, the interfacial phase shift can be largely deviated from the two values, even leading to very large value close to 2π . Mathematically, the singularity point as the reviewer mentioned is reasonable where the interfacial phase shift is overwhelmingly dominant over the propagation phase accumulation. Because of the singularity, the resonance wavelength becomes extremely sensitive to the infinitesimal thickness of the dielectric film when the total interfacial phase shift is infinitely approaching 2π . It is the kernel of the design principle of this work that the ultra-sensitivity of layer-dependent resonances is optimized based on the unique dispersion of 2D layered MoS₂ film deposited on the gold substrate.

Please see revisions in Page 6.

Comment 2) The theory behind the laser exfoliation process needs to be clarified more in detail. The depth of focus of the focused laser beam will cover the entire film thicknesses, but the authors seem to suggest that heating occurs from the top layer. I believe that the gold substrate plays a significant role in the thermal heating process and the overall thermal profile of the film that the authors are describing, but its role is not mentioned in the manuscript.

Reply: We thank the reviewer for constructive comments. We have taken the reviewer's suggestion and modelled the heat diffusion process. Three-dimensional temperature distribution of the layered MoS₂ film can be numerically obtained through thermal diffusion equations (see Supplementary Note 2). In the model, MoS₂ film exhibiting anisotropic thermal diffusion coefficients with 43 mw/(nm*k) and 0.43 mw/(nm*k) in the lateral and axial directions, respectively, is adopted. The gold substrate has a thermal conductivity of 320 mw/(nm*k) and SiO₂ substrate has a thermal conductivity of 0.11 mw/(nm*k). For comparison, Fig. R1 shows calculated the temperature distribution within the MoS₂ layer deposited on the Au and SiO₂ substrate after being irradiated by a CW laser beam by using numerical method. The temperature distribution $U_i(t, r)$ of layer number of 15 at time t and off-center position r using thermal diffusion equations is calculated. The laser beam has a Gaussian profile. It is revealed that the axial temperature gradient in the MoS₂/Au sample is much sharper than that in MoS₂/SiO₂ sample. Au has a thermal conductivity 300 times larger than the SiO₂ substrate, which results in a quick thermal dissipation to the environment. The presence of gold substrate causes a significant temperature drop in the bottom layers, which helps to laser exfoliate the atop atomic layers without adversely affecting the bottom atomic layers. This forms the basis for the layer-by-layer laser exfoliation with atomic accuracy.

Please see revision in the main text in Page 7, Supplementary Note 2 and Figure S7.

Fig. R1. The calculated temperature distribution in the xz plane within a 10 nm thick MoS_2 film deposited on the Au (a) and SiO_2 substrate (b). The diameter of diffraction-limited focal spot is set 430 nm by an objective lens with numerical aperture of 0.75 and laser power of 1.5 mW are used in the numerical calculation.

Comment 3) The laser power density, not the absolute power, should be reported. A discussion of the damage to the film should be included as this can significantly modify the optical properties of the film.

Reply: We have followed the reviewer's suggestion. The laser power has been converted into power density. The dynamic range of the laser power density has been given in Figs. 2b and c. Once the power density exceeds $13 \text{ mW}/\mu\text{m}^2$, laser exfoliation occurs with a minimum thickness step of 3 nm, corresponding to 4 atomic layers. Once the power density exceeding $40 \text{ mW}/\mu\text{m}^2$, complete removal of the atop MoS_2 layer takes place as indicated by both Raman scattering image and AFM image. When the power density is below $12 \text{ mW}/\mu\text{m}^2$, no notable optical property change can take place as evidenced by Raman scattering before and after irradiances.

Please see discussions in revised text in Page 8.

Comment 4) The captions and figures require improved labeling in the Supplementary section. For example, the 'efficiency' labeled in Figure S6 needs to be specified. Which efficiency is this?

Reply: We thank the reviewer for careful reading and constructive comments to improve the readability. In Figure S6, the efficiency is to characterize the diffraction efficiency defined as the ratio between the first order diffraction intensity and the reflected intensity by summing up both 0th order and the first order intensities.

#####

Reviewer #2 (Remarks to the Author):

The manuscript used laser ablation/sublimation to reduce thickness of a bulk MoS₂ layer to change Fabry-Perot resonance to change color of the film. FP resonance based color generation is most widely known and used phenomenon. Many materials are patterned following conventional lithographic techniques as well as lasers to form FP color. There are many publications on this topic. My few specific questions/comments are:

Reply: We thank the reviewer for his or her time for reviewing our manuscript. However, we completely disagree with this reviewer for treating our work as conventional Fabry-Perot resonances by totally ignoring the fundamental difference in mechanisms and potential impacts.

As depicted by a universal physical model in Eq. 1, contributions from both propagation phase accumulation (thickness dependent) and interfacial phase shifts (loss dependent) govern the resonance. The conventional FP resonators, which are dominant by propagation phase accumulation in lossless dielectrics, require a large thickness of multiple integers of $\lambda/4$. In stark contrast, the resonance in our configuration consisting of layered MoS₂ film deposited on the gold substrate is overwhelmingly dominant by the interfacial phase shifts. Therefore, it alleviates the thickness of the resonator to as small as a few atomic layers, which is out-of-reach by conventional FP resonators and makes it fundamentally different.

Because of fundamentally different mechanisms, the interfacial phase shifts are less sensitive to incident angles compared to the case in traditional FP cavities where the propagation phase accumulation is dominant and sensitive to incident angles. Both simulation and experimental results demonstrate an angle robust performance as shown in Supplementary Fig. S10.

This large interfacial phase shift leads to an ultra-sensitivity in resonance wavelength to atomically thin layers for both vivid color appearances and strong reflectance amplitude modulation for diffractive stereoscopic images. Conventionally, to demonstrate wide color gamut traditional FP cavities require a large variation in thicknesses with surface corrugations on the order of a few tens to hundreds of nm. Because of the ultra-sensitivity by our approach, it opens the possibility for nanometric flat optics with surface corrugations less than 15 nm. Moreover, our method can extend the potentials of 2D materials to be used in more practical applications including color images, displays and diffractive meta-optics.

Comment 1) What is the film quality of the PLD grown MoS₂ layer? It will have many defects and will be a poor polycrystalline film. I don't see any justification for calling it a 2D material in this context. Here the laser is doing bulk micromachining of the material.

Reply: Pulsed laser deposition of MoS₂ has been widely reported in previous literatures. For example, quasi-epitaxial MoS₂ thin films with (002) preferred orientation have been observed for films grown on GaN (APL, 106, 052101, 2015). Layered growth of MoS₂ thin films has been reported on Al₂O₃, SiO₂ and HfO₂ substrates, (ACS Nano, 10, 6054, 2016, J. Mater. Chem. C, 7, 2522, 2019), metal Ag substrates (ACS Appl. Mater. & Interf., 6, 15966, 2014). They all share similar structure of 2D layered, crystallized structures and (002) textured orientations with Van der Waals force between layers along the surface normal orientation. In the revised manuscript, we characterized the crystal structure of the deposited MoS₂ thin film. We show the XRD, and TEM characterization results of our samples in Figs. S1 and S2. We observe exactly the same (002) textured growth of 2H- MoS₂ by XRD, suggesting a layered structure stacking along the surface normal orientation. From the XRD pattern, we can

calculate the (002) lattice spacing from the Bragg's law to be 0.626 nm.

We also observed layered crystal structure of MoS₂ from the STEM and HR-STEM images, as shown in Fig. S2. Fig. S2(a) shows the STEM image of the sample fabricated by focused ion beam (FIB) milling. The as-prepared MoS₂ thin film was ~40 nm thick. A capping SiO₂ layer was deposited by sputtering to protect the MoS₂ film before FIB. The zoom-in figure of the MoS₂ thin film measured by HR-STEM is shown in Fig. S2(b). The sample showed clear layered structure of the MoS₂ lattice. We measured the layer spacings from the line profile of the intensity indicated in Fig. S2(b). The measured spacing between two layers are 1.315 nm, indicating an interlayer distance of 0.657 nm. This value matches exactly with XRD characterization results. Notice this number also matches with the interlayer spacing value of MoS₂ fabricated by other methods.(*Inorganic Chemistry Frontiers* **5**, 3099-3105 (2018), *Materials Today* **20**, 83-91 (2017))

Fig. R2 X-ray diffraction pattern of a 10 nm MoS₂ thin film deposited on SiO₂/Si by pulsed laser deposition

From the above characterization results, we can summarize the MoS₂ crystal structure as follows. The deposited MoS₂ crystallized into the 2H phase showing (002) texture. The crystals show layered structures from the HR-STEM image. From these results we are confident to say this is a 2D material like MoS₂ fabricated using other methods.

Of course, this material is not a single crystal in the in-plane orientation. The textured grains may still rotate in-plane, but sharing the (002) orientation along the surface normal orientation. This is similar to CVD grown MoS₂ thin films. Indeed, the PLD grown MoS₂ may show more point defects such as S vacancies and grain boundaries compared to single crystal MoS₂. However, this did not show a strong influence on the material's optical constants. We characterized the material's optical constants by spectroscopic ellipsometry, which matched well with literature reports on either single crystal MoS₂ or deposited MoS₂ thin films, as will be discussed in detail in the next question. In addition, the layered structures are mostly stacked along the out-of-plane direction, making the micromachining possible. The thinning down process was achieved by laser exfoliating the material layer by layer due to the weak Van der Waals force between layers. On the other hand, micromachining a bulk material in the sub-nanometer scale would be very difficult if the material were showing 3D bonding structures. Therefore, the grown MoS₂ with layered 2D

material structure made our experiments possible, which was also one of the critical innovation points of our work. Please see changes in Supplementary Figs. S1 and S2.

Fig. R3 (a) STEM image of a MoS₂ thin film grown on SiO₂ (b) HR-STEM imaging showing layered structure of the MoS₂ thin film. (c) Intensity profile of along the cross-section shown in (b).

Detail of the STEM Characterization Methods

The sample is a cross section sample cut with FIB(FEI Quanta 3D FEG). High-resolution ADF-STEM were performed in a probe-corrected STEM (FEI Titan Chemi STEM) operated at 200 kV. For illumination and *in-situ* fabricator, a convergence angle of 21.4 mrad , a probe current of ~ 70 pA , a range of acceptance angle of ADF detector of 43.4–200 mrad , and a pixel dwell time of 10 μ s were used for the characterization.

Comment 2) The manuscript will benefit if authors show film characterizations, electrical as well as optical.

Reply: We thank the reviewer for the comments to further improve our manuscript. In Fig. S3(a) we show 2 cm by 2 cm sized as-grown MoS₂ thin film by PLD on a SiO₂/Si substrate. We measured material's optical constants using spectroscopic ellipsometry (J. A. Woollam, RC2) from the ultraviolet to near infrared wavelength range, as shown in Fig.R4(b). The spectroscopic ellipsometry data were analyzed using the CompleteEASE software. The film thickness is 9.05 nm. Two absorption peaks were observed at photon energies of 2.0 eV and 2.9 eV, corresponding to A/B exciton absorption and C/D exciton absorption peaks respectively, indicating the deposited MoS₂ show similar band structure compared to CVD grown MoS₂ thin films. (APL, 104, 013114, 2014). The index of refraction and extinction coefficient also quantitatively agree with CVD grown MoS₂ thin films. This result proves that the PLD grown MoS₂ show similar optical properties compared to CVD grown thin films.

We did not characterize the electrical properties of the deposited MoS₂. It is likely that due to defects such as S vacancies, this material may show lower resistance compared to single crystalline thin films. In terms of relevant properties to our device, these effects will all contribute to the optical constants, which can be described by the Drude-Lorentz model. Since the measured n,k values match with MoS₂ fabricated by other methods, we do not see a strong influence from the different electrical properties to the optical constants. Therefore, we believe the optical constants contain all the relevant material properties that governs the experimental results.

Fig. R4 (a) Photograph of MoS₂ thin film deposited by PLD on a 2 cm by 2 cm SiO₂/Si substrate. (b) Optical constants of MoS₂ in the visible to near infrared wavelength range

Comment 3) Angular dependency of the color should be studied for any useful application.

Reply: We have followed the reviewer's suggestion. The reflection spectra as a function of wavelengths and incident angles are calculated and illustrated in Fig. R5. Because the ultra-sensitive resonance stems from interfacial phase shifts which are immune to the incident angle, the color images demonstrate an angle robust performance from normal to 60° oblique incidences. The related discussion has been added in Page 10 and Supplementary Fig. S10.

Figure R5. Angular dependency of reflectivity spectra. (a,b). The calculated spectra for s- and p-polarization, respectively, for angles of incidence from 0° to 80° for an Au film coated with 10nm of MoS₂(the value of reflectivity is indicated by the colour bars). (c) Experimental results of angle robust performance of color prints with incident angles from 10° to 60°

#####

Reviewer #3 (Remarks to the Author):

This manuscript describes a simple and efficient method to exfoliate MoS₂ and also shows fidelity color images and a binocular stereoscopic image by irradiating continuous laser to multi-layer MoS₂ on the Au substrate as applications. Indeed, I believe this method can extend the potentials of 2D materials to be used in more practical applications. The manuscript is clearly written, contains noble results, and deserves to be acceptable in Nature Communications. However, I feel that some minor points in the manuscript need to be examined.

Reply: We thank the reviewer for fully endorsing the novelty and impact of our work, and supporting its publication after revisions.

Comment 1) It is not explained why the gold substrate is better than aluminum and silicon substrate. Because of material loss? or plasma frequency? Is it normal for any TMD materials? Please explain shortly why.

Reply: We thank the reviewer for his or her professional comments. The choice of gold substrate is simply because that the total interfacial phase shift for MoS₂/Au sample is near 2π , which can bring a large sensitivity of the resonance wavelength to the MoS₂ thickness. For detailed comparison, please refer to the Supplementary Fig. S5. Clearly, it shows that the interfacial phase shifts for MoS₂/Au sample is superb among the other substrates such as silicon, Al and Ag. It should be noted that the design principle can be generalized for other materials as well. For different TMD materials, the optical constants disperse at different wavelengths, indicating that the best matching substrate may change from one material to the other materials.

Comment 2) Figure 1c is not easy to understand. What is the main purpose of this graph? To notice broadband interfacial phase shifts? Where is the region 2π phase shifts? The red color region? or the phase is abruptly jumped (where n is 2, k is 3.5)? What is the k in Figure 1c? Extinction coefficient?

Reply: The purpose of Fig. 1c is to illustrate the broadband total interfacial phase shift of MoS₂/Au configuration close to 2π . The 2D color plot shows the calculated interfacial phase shift of lossy dielectric thin films with different complex refractive indices (n for real parts and k for imaginary parts) deposited on Au substrate at the wavelength of 532 nm. The red(blue) color indicates regions exhibiting the largest(smallest) interfacial phase shifts close to $2\pi(0)$. To better visualize the sensitivity of different materials, the dispersions of Ge, Graphene and MoS₂ at visible ranges from 300 nm to 800 nm are superimposed. Clearly, the dispersion of MoS₂ covers the widest region with total interfacial phase shifts close to 2π (red). To avoid confusion, we have made clarifications in Page 7 and figure captions.

Comment 3) It is not distinguishable well between MoS₂ and graphene, and Ge and background phase color due to similar colors in Figure 1c. Please change the colors to distinguish well. I cannot understand why the calculated results of Ge and graphene are suddenly shown without having an explanation. These results make me confused as well.

Reply: We apologize for the confusion it caused. The color schemes are modified to better distinguish between these curves. The configuration of MoS₂/Au sample is optimized for ultra-sensitivity to the thickness of the lossy dielectric layer. The resonance sensitivity and associated spectra tunability are overwhelmingly ascribed to the interfacial phase shifts, which strongly depend on the wavelength-dispersive optical constants of the lossy dielectric layer. For comparison, the results of Ge and graphene are added to illustrate the large and

broadband interfacial phase shift induced sensitivity. To avoid confusion, we have made clarifications in Page 7 and figure S6 captions.

Comment 4) The authors mentioned and compared the sensitivity of graphene, Ge, and MoS₂. However, it is not clear the definition of sensitivity and how the number (12.83 nm) could be obtained. Please explain more details to be understandable.

Reply: We have followed the reviewer's suggestion and explained the calculation of the sensitivity of MoS₂. The sensitivity is defined as the wavelength shift of resonances normalized to the number of material layers. Please see revisions in Page 7.

Comment 5) Line 96,137, 166, and 208, "showcases"  shows?

Reply: We thank the reviewer for careful reading. The sentence has been corrected.

Comment 6) Figure 2 describes the exfoliated MoS₂ film thickness depend on laser power. However, the exposure time of the laser is not explained.

Reply: We thank the reviewer for professional comments. Since we are using continuous laser with a scanning positioner to control the irradiation condition, the effective exposure time can be evaluated by dividing the diameter of laser focal spot by the scanning velocity, which yields an equivalent exposure time of 4.33 ms. Please see revisions in Page 13.

REVIEWER COMMENTS

Reviewer #1 (Remarks to the Author):

In their response letter, the authors argue that their use of the layer-by-layer laser exfoliation method differentiates their work from that of M.A.Kats et al. Nature Materials (2013) paper. I can see their argument, but if this is truly to be a point of novelty, its explanation and analysis need to be very clear and robust. Although the authors have added material in response to my previous comment on this issue, I am having some difficulty following their explanation, and hope that this can be clarified.

First, the schematics that they use in Fig1a and Fig S7 give the wrong impression. The focused light (or point spread function) cannot be as small as the size of an atom.

In line with this, the authors use a Beer's Lambert-type law to model the intensity profile of the light inside the MoS₂ film in their analysis of the heat diffusion. This is not an appropriate model to use since the Beer's-Lambert law applies to systems with sizes much larger than the wavelength. As the authors explain in detail in the text, the thin MoS₂ thin film can show resonances even at ultra-subwavelength thicknesses. In general, nanostructures including thin films support resonant modal profiles rather than simple exponentially attenuating profile. This model needs to be robust in order to support their claims of a 120 degrees/ nm temperature gradient, which is quite remarkable if true.

Reviewer #2 (Remarks to the Author):

The manuscript is only incremental compared to the 2013 Nature Materials paper by M.A. Kats et al. on nanometer scale FP cavities.

The paper will benefit from comparing with other structural color papers in terms of color gamut, color brightness, angle independency etc parameters.

I believe this paper is better suited for another journal focused on fabrication.

Reviewer #3 (Remarks to the Author):

This manuscript describes a simple and efficient method to exfoliate MoS₂ and also shows fidelity color images and a binocular stereoscopic image by irradiating continuous laser to multi-layer MoS₂ on the Au substrate as applications. The revised manuscript is well prepared and clearly written. Especially, Authors clearly answered my question. In this reviewer's view, the manuscript is acceptable for publication in Nature Communications.

#####

Reviewer #1 (Remarks to the Author):

In their response letter, the authors argue that their use of the layer-by-layer laser exfoliation method differentiates their work from that of M.A.Kats et al. Nature Materials (2013) paper. I can see their argument, but if this is truly to be a point of novelty, its explanation and analysis need to be very clear and robust. Although the authors have added material in response to my previous comment on this issue, I am having some difficulty following their explanation, and hope that this can be clarified. First, the schematics that they use in Fig1a and Fig S7 give the wrong impression. The focused light (or point spread function) cannot be as small as the size of an atom. In line with this, the authors use a Beer's Lambert-type law to model the intensity profile of the light inside the MoS₂ film in their analysis of the heat diffusion. This is not an appropriate model to use since the Beers-Lambert law applies to systems with sizes much larger than the wavelength. As the authors explain in detail in the text, the thin MoS₂ thin film can show resonances even at ultra-subwavelength thicknesses. In general, nanostructures including thin films support resonant modal profiles rather than simple exponentially attenuating profile. This model needs to be robust in order to support their claims of a 120 degrees/ nm temperature gradient, which is quite remarkable if true.

Reply: We thank the reviewer for his/her valuable comments. In view of the reviewer's comments, we make the following modifications.

1) For clarity, we have modified the focused laser spot size of the schematic diagram in Figure 1a and Figure S7.

2) We have followed the reviewer's suggestion and replaced the model using the Beers-Lambert law with full field simulation. By using the finite difference time domain (FDTD) method, we have simulated the field distribution within the MoS₂ layers with a thickness of 10 nm upon focused laser beam illumination. The simulated field distribution and its cross-section plot along the stacking layer direction are shown in Fig. R1c and d. Then the absorbed optical power flow at each point was extracted and used to calculate the temperature distribution by solving the thermal diffusion equations. The updated results are added in Figure S7, and shown in Figure R1. Compared with the model using the Beers-Lambert law, the model using simulated field distribution is more accurate and unveils a temperature gradient up to 74.4 degrees per nm along the stacking layers.

3) Apart from the low emissivity of atomically-thin MoS₂ layers, the available thermal camera has limited resolutions close to sub mm that prevents us from direct observation of such temperature gradients within the focal point. Nevertheless, experimental results in Figure 2 in the main text unambiguously prove that layer-by-layer laser exfoliation can be achieved by controlling the laser powers, although the laser focus is so much larger than the thickness of the MoS₂ nanometric film. It can be inferred that the temperature gradient due to the anisotropic thermal conductivity plays an important role for the observed nanometric laser exfoliation. To directly mapping the temperature gradient, it is subject to further study in the following work.

Fig. R1. Calculated temperature distribution during laser exfoliation. (a) Schematic diagram of laser thinning, K_i and K_o represents in-plane and out-of-plane thermal conductivity, respectively, and K_{sub} represents thermal conductivity between MoS₂ and substrate. (b) The calculated temperature distribution in the XZ plane where the absorbed optical power flow at each point was extracted by the FDTD simulation. (c) The field distribution within the MoS₂ layers upon focused laser beam illumination. The amplitude is 1.5 mV/m. The black dotted line indicates the MoS₂ layer. (d) The electric field intensity at the white dotted line in (c). (e) The vertical direction plots of temperature spatial distributions, showing a sharp temperature gradient up to 74.4 degrees per nm along the stacking layers.

#####

Reviewer #2 (Remarks to the Author):

The manuscript is only incremental compared to the 2013 Nature Materials paper by M.A. Kats et al. on nanometer scale FP cavities.

The paper will benefit from comparing with other structural color papers in terms of color gamut, color brightness, angle independency etc. parameters.

I believe this paper is better suited for another journal focused on fabrication.

Reply: We thank the reviewers for re-reviewing our revised manuscript and offering further comments. Although our work and the work reported in 2013 Nature Materials share the interfacial phase shifts and the consequent color effect in common, we reinforce that the innovation of this work making it different from the referenced NM paper lies in the following facts.

- 1) Firstly, the anisotropic thermal diffusion coefficients of layered MoS₂ films enables the layer-by-layer laser exfoliation method for post-fabrication laser prints with atomic accuracy. Even the initial thickness of the layered film of 20 nm that is within the focal depth of the tightly focused beam of 1,324 nm, it allows exfoliation of atop atomic layers without adversely affecting the bottom layers. Thanks to the orders-of-magnitude difference in thermal diffusion coefficients in the lateral and vertical directions, a sharp temperature gradient of 74.4 degrees/nm can be achieved to allow laser exfoliate with a demonstrated thickness step of 3 nm corresponding to 4 atomic layers at each laser sweeping. This top-down laser writing method with atomic accuracy opens the possibility for in-situ thickness control for laser prints even after fabrication, which makes it fundamentally different from the case where isotropic Ge thin film was prepared in lithography without the post-fabrication capability (Nature Materials, 2013).
- 2) Secondly, we report the first demonstration of diffractive laser prints for binocular stereoscopic image based on such strong amplitude modulations. The ultra-sensitive layer-dependent resonances stemmed from interfacial phase shifts cannot only lead to vivid color appearances but also large reflectance amplitude modulations of 0.44 with a surface corrugation of merely 15 nm. Consequently, two sets of nanometric flat diffractive gratings have been printed for binocular stereoscopic images, which opens the potential for planar diffractive optics, that is different from and out of scope of the NM paper.

In addition, we have followed the reviewer's suggestion and compared the performance with other structural color papers. Please see supplementary Table S2.

#####

Reviewer #3 (Remarks to the Author):

This manuscript describes a simple and efficient method to exfoliate MoS₂ and also shows fidelity color images and a binocular stereoscopic image by irradiating continuous laser to multi-layer MoS₂ on the Au substrate as applications. The revised manuscript is well prepared and clearly written. Especially, Authors clearly answered my question. In this reviewer's view, the manuscript is acceptable for publication in Nature Communications.

Reply: We thank the reviewer for fully endorsing the novelty and impact of our work, and supporting its publication.

REVIEWERS' COMMENTS

Reviewer #1 (Remarks to the Author):

The authors have redone the simulations on laser-induced heat generation to address my concerns. They have modified the temperature gradient accordingly. However, my original concern with the schematics (Fig 1a and Fig S7a) still has not been sufficiently addressed. The authors used a 0.75 NA objective lens with 532nm light focus light and exfoliate the MoS2 multilayers. The focused light is diffraction limited and cannot be on the order of the few-angstroms that appear in the schematics. The authors need to reconsider the schematics to avoid giving the wrong impression.

#####

Reviewer #1 (Remarks to the Author):

The authors have redone the simulations on laser-induced heat generation to address my concerns. They have modified the temperature gradient accordingly. However, my original concern with the schematics (Fig 1a and Fig S7a) still has not been sufficiently addressed. The authors used a 0.75 NA objective lens with 532nm light focus light and exfoliate the MoS2 multilayers. The focused light is diffraction limited and cannot be on the order of the few-angstroms that appear in the schematics. The authors need to reconsider the schematics to avoid giving the wrong impression.

Reply: We thank the reviewer for his/her valuable comments. In view of the reviewer's comments, we have made the following modifications.

For clarity, we have modified the focused laser spot size in the schematic diagram in Figure 1a and Figure S7. In detail, to avoid the misunderstanding that the laser can focus down to only a few atoms, we increase the size of the laser spot that is illustrated as being far larger than the MoS2 atoms in the schematics and moderate the in-plane laser density gradient. We hope these modifications can address the reviewer's concerns.